# Interfacial Bond Behavior of High Strength Concrete Filled Steel Tube after Exposure to Elevated Temperatures and Cooled by Fire Hydrant

**DOI:** 10.3390/ma13010150

**Published:** 2019-12-31

**Authors:** Zongping Chen, Jiyu Tang, Xingyu Zhou, Ji Zhou, Jianjia Chen

**Affiliations:** 1College of Civil Engineering and Architecture, Guangxi University, Nanning 530004, China; tangjiyu001457@163.com (J.T.); zh-xy-59@139.com (X.Z.); 17687657937@163.com (J.Z.); jianjiaata@163.com (J.C.); 2Key Laboratory of Disaster Prevention and Structure Safety of Chinese Ministry of Education, Guangxi University, Nanning 530004, China

**Keywords:** elevated temperatures and water-cooling exposure, high-strength concrete, circular steel tube, bond strength, slip behavior

## Abstract

For the engineering structure in case of fire, a fire hydrant is generally used for extinguishing the fire. This paper presents an experimental investigation on interfacial bond behavior of high-strength concrete-filled steel tube (HSCFST) after exposure to elevated temperatures and cooled by fire hydrant using the pull-out test of 22 specimens. According to the experimental study, the failure mechanism of HSCFST exposed to elevated temperatures and water-cooling (ETWC) was revealed, the influence of various parameters on the bond behavior was analyzed, and the calculation formula of the bond strength of HSCFST subjected to ETWC was put forward. The results show that the load-slip curves of the loading end and the free end of the specimen are basically similar, and can be divided into three types of typical curves. In the push out test, the strain on the outer surface of the steel tube is exponentially distributed with its distance from the loading end. After ETWC exposure, the bond strength of the specimen is less affected by the concrete strength, which is inversely proportional to the anchorage length, and it is basically stable after the constant temperature duration is longer than 60 min. With the increase of the maximum temperature, the ultimate bond strength increases first, then decreases and then increases, and the residual bond strength increases first and then decreases. Besides, the study indicate that cooling method has significant influence on the bond behavior, compared with natural cooling specimens, the ultimate bond strength, residual bond strength, and shear bond stiffness of water-cooling specimens are smaller, and the interfacial energy dissipation capacity is larger.

## 1. Introduction

High-strength concrete filled steel tube (HSCFST) is a kind of structural component which is filled with high strength concrete in the steel tube, and the steel tube and the inner core concrete bear the external load together. The application of HSCFST in high-rise building and long-span bridge is the most effective and economic structural form [1], which has high bearing capacity, large stiffness, and excellent seismic performance [2,3,4]. Therefore, the research on the mechanical properties of HSCFST structure has important theoretical significance and engineering application value. Good bonding performance between the steel tube and core concrete, is the important foundation of the two work together as a whole. The studies have mostly failed to take into account the bond-slip between steel and concrete. And in practical engineering, especially in some nodes or special parts, the HSCFST members may only bear the stress of the steel tube or the core concrete. In this case, studying the bond-slip behavior of HSCFST is very necessary. Virdi et al. [5] and Shakir Khalil et al. [6] started to conduct research through push-out test at the earliest time. At present, this test method is mostly used in researches on the bond-slip of HSCFST. Research on the bond behavior of HSCFST interface under normal temperature has always been the focus of scholars. Cameron et al. [7] conducted 20 push-out tests on circular HSCFST columns with a maximum diameter of 610 mm. The obtained test results showed that with large diameter tubes with large d/t ratios, shrinkage could lead to very little bond stress capacities. Small diameter tubes with smaller d/t ratios developed large bond stress capacities. Xu et al. [8] investigated the bond carrying capacities of 17 short, pre-stressing HSCFST columns by means of expansive cement and three short, conventional HSCFST columns, and they found that both concrete mixes and dimensions of the steel tube had important influence on expansive behaviors of pre-stressing HSCFST specimens. Generally, pre-stress obviously increases before 15-day age and almost keeps in constant after 25-day age. Aly et al. [9] tested 14 HSCFST circular column specimens to study the bond behavior of composite columns subjected to static loading protocols, the authors concluded that the interface bond strength in HSCFST circular columns with normal strength concrete is found to be higher than that with high strength concrete. Bond mechanisms in rectangular HSCFST columns, and the relative contribution of each component were studied by Qu et al. [10], who found that macrolocking was the main mechanism in rectangular HSCFST columns contributing to the bond strength followed by friction.

With the wide application of HSCFST structure system in practical engineering, it is more and more important to consider the occurrence of accidental fire, so the bond behavior of HSCFST exposure to fire has gradually attracted attention. Previous studies have shown that some physical and chemical changes can occur in the steel and concrete materials during fire exposure [11,12,13,14]. Therefore, it is expected that the fire exposure will lead to the internal force redistribution in a HSCFST column and possible deterioration of the bond strength. Chen et al. [15,16] carried out experimental study on the bond behavior of HSCFST exposure to high temperature, the authors concluded that with the increase of the maximum temperature, the interfacial bond strength of HSCFST increased first and then decreased, and the bond strength was inversely proportional to the anchorage length. Tao et al. [17] conducted 64 push-out tests on HSCFST columns, which had been exposed to fire for 90 min or 180 min, respectively. The test results indicated that fire exposure had a significant effect on the bond between a steel tube and its concrete core. Song et al. [18] conducted and tested 28 HSCFST specimens, including 12 reference specimens at ambient temperature and 16 post-fire specimens, to investigate bond behavior of HSCFST columns subjected to fire. The results indicated that the bond of specimens with normal interface could be completely broken in the fire.

For engineering structures, when the fire occurs, fire hydrant is generally adopted to extinguish the fire. Cooled by a fire hydrant will make the temperature of the surface of the building structure drop rapidly in a short time, which may affect the performance of the structure. There is no research on the mechanical properties of HSCFST members subjected to elevated temperatures and water-cooling (ETWC), however, considering the influence of the whole process of fire on bond behaviors of HSCFST, this paper discusses the interfacial bond behaviors of HSCFST subjected to ETWC, which can provide theoretical basis for the research and analysis of the mechanical properties of HSCFST subjected to ETWC.

## 2. Experimental Program

### 2.1. Specimen Design and Preparation

In order to study the bond-slip behavior of HSCFST exposure to ETWC, 22 specimens were designed to carry out the push-out test. Five parameters were considered in this test: (a) Maximum temperature (i.e., 200 °C, 400 °C, 600 °C, 800 °C); (b) design value of concrete strength (i.e., 60 MPa, 70 MPa, C80 MPa); (c) constant temperature duration (i.e., 30 min, 60 min, 90 min); (d) cooling method (i.e., water-cooling, natural cooling); (e) anchorage length (i.e., 250 mm, 400 mm). Moreover, there was also an unheated specimen left at room temperature (20 °C) as a contrast. Detailed design and measured parameters of the specimens are given in Table 1 and the geometric dimensions and structure of the specimens are shown in Figure 1. In Table 1, the specimens were named according to the following rules: (a) HSCFST represents high-strength concrete filled steel tube specimens; (b) the No.1–22 represents the number of the specimens; (c) W and N respectively represent that the specimen is cooled by water and cooled naturally. Taking HSCFST-2W as an example, the number indicates that the specimen is NO.2 HSCFST specimen subjected to ETWC. And as shown in Table 1, *T* is the maximum temperature; *t* is the constant temperature duration; *H* is the total length of the specimen; *l_a_* is the measured anchorage length.

### 2.2. Material Properties

The concrete mix proportion is given in Table 2. P.O 42.5 ordinary Portland cement, well-graded small and medium-size river sand, urban tap water and gravel of 5~25 mm diameter were used in the test, first-grade fly ash as well as silica fume were used as admixture, and high range polycarboxylate water reducer from concrete mixing station was used. The physical properties of sand and coarse aggregate are shown in Table 3 and Table 4, tested according to the GB/T14684-2011 [19] and the GB/T14685-2011 [20] respectively. The steel tube adopted the low carbon steel Q235 circular steel tube, the weld was straight weld, the outer diameter of that was 165 mm, and its wall thickness was 4 mm.

### 2.3. High Temperature Treatment and Cooling Method

The change diagram of the ambient temperature of the specimen under high temperature treatment and different cooling method is shown in the Figure 2. It can be seen from the figure that water-cooling makes the temperature drop faster. The specimens can be subjected to high temperature treatment and be exposed to fire on its four surfaces sides in a RX_3_-45-9 electric oven. With its own heating temperature control system and temperature sensor, specimens rose to the target temperature at a steady rate in the furnace, and then kept the temperature in the furnace unchanged within the design time. After that, some parts of specimens were cooled by water and the remaining specimens were naturally cooled in the furnace.

The specimens, to be water-cooling, were transported from the high temperature furnace to the handcart flush with the high temperature furnace mouth, and then moved quickly to the outdoor for simulating fire sprinkler treatment. A single fire hydrant was used to cool the specimen within 5 m. The water consumption of the spray gun was recorded and controlled by the water meter, and the specimen was sprayed with water for 25 min. According to the two Chinese codes, namely GB50016-2014 [21] and GB50974-2014 [22], respectively, the diameter of hydrant port used in the test is 65 mm, and the diameter of monitor nozzle is about 19–20 mm. Before water-cooling, the water output of a single spray gun can be kept at about 15 L/s by using a water meter and a stopwatch, and then the water spray treatment can be carried out. The field operation of the test is shown in Figure 3. After water-cooling, the specimen was put into a static position, and its mass subjected to water-cooling for the first, second, and third days and before loading were recorded respectively, so as to obtain its mass loss rate.

### 2.4. Test Setup and Loading Process

After the specimens were subjected to ETWC, the push-out test was carried out. In order to obtain the load-slip curve of the whole process of the test, the test was under displacement control, and the loading rate was 0.2 mm/min. When the slip at the free end reached about 10 mm and there was no significant change in the load, at this time, the test was regarded as finished. The test setup used to conduct the push-out tests is shown in Figure 4. The bottom of the specimens was the loading end of the test, and the concrete core was pushed out from the bottom to the top through the thick steel block, which had a cross-section slightly smaller than the inner diameter of circular steel tube. Two displacement meters were used to record the slip of the core concrete at the loading end and the free end during the loading process. Wherein, No.1 displacement meter was located on the steel block with high stiffness at the bottom to measure the slip at the loading end. No.2 displacement meter was placed on the steel block which was poured together with the core concrete to measure the slip at the free end. In order to eliminate error of the LVDTs, the load is applied to 10% of the estimated ultimate load before the formal loading of each test, and then the unloading is carried out to ensure the normal operation of the instrument. At the same time, the strain gauges were installed along the length of the steel tube to record the strain distribution along the anchorage height. The specific arrangement of strain gauges is shown in Figure 1.

## 3. Experiment Results and Analysis

### 3.1. Phenomenon of Water-Cooling Process

Through the observation during the process of spraying water, it was found that after different high temperatures, the phenomena of each specimen showed obvious differences during the water-cooling test. After exposure to temperature at 200 °C, no obvious water vapor was generated on the surface of the specimen when spraying water, and the temperature of the specimen after spraying water was close to the ambient temperature. After exposure to temperature at 400 °C, a small amount of water vapor was generated on the surface of the specimen within 5 min after spraying water. After the water-cooling, the specimen still had waste heat, and the temperature of the external steel tube was higher than that of the concrete core. After exposure to temperature at 600 °C, a large amount of water vapor was generated during the process of water-cooling. At this time, the core concrete was accompanied by a slight cracking sound, and the generation of water vapor was not obvious after 10 min. After spraying water, the temperature of the specimen was high, so it was not allowed to touch directly and the temperature of the specimen fell to room temperature for about 6 h. At 800 °C, not only was there a lot of water vapor generated in the process of water-cooling, but the core concrete gave out a violent cracking sound. During the whole process of spraying water, the water vapor generated continuously. After the water spraying, the water remaining on the surface of the specimen evaporated quickly, and the core concrete showed obvious cracking.

### 3.2. Phenomenon after Elevated Temperatures and Water-Cooling

Compared with the specimens at room temperature, the specimens subjected to ETWC underwent different physical and chemical changes. For the test blocks, their specific phenomena are as follows: at room temperature, the test blocks were cyan with no cracks on the surface. At 200 °C, the test blocks became light blue with a small number of micro cracks on the surface. At 400 °C, the test blocks were yellowish with more micro cracks on the surface. At 600 °C, there were obvious burning marks on the surface of the test blocks, with dark color and obvious cracks. At 800 °C, the test blocks were white and the surface cracked and the coarse aggregate was exposed. It can be seen that the ETWC exposure causes these physical changes in the test blocks. For the specimens, the surface color of the steel tubes changed noticeably and the formation and peeling of the oxide layer were obvious. At room temperature, the specimens were light brown. When *T* = 200 °C, the surface color of the specimens was brownish yellow. When *T* = 400 °C, the outer steel tubes of the specimens were covered by the oxide layer and turned yellow-brown. When *T* = 600 °C, the oxidation degree of steel tubes was higher, the color became reddish brown, and the surface was accompanied by transverse lines. After exposed to 800 °C, the oxide layer on the outer surface of the steel tube partially fell off, and the color was black. After various elevated temperatures and water-cooling, the apparent morphology of each specimen is shown in Figure 5.

### 3.3. Mass Loss Rate

After elevated temperatures, free water and bond water in core concrete partially or completely evaporated, resulting in mass loss. However, in the process of water-cooling, high-pressure water or water vapor aggravates the development of cracks and penetrates into the interior of concrete, where hydration occurred again, thus increasing its weight. Because of the compactness of the internal structure of high-strength concrete, the concrete may also burst under the action of temperature changes and the pressure of water vapor, resulting in its mass loss. After water-cooling, the interior temperature of concrete was still very high, and the water may further hydrate. At the same time, the cracks in the concrete surface were also conducive to the re-evaporation of water into the concrete interior. Therefore, in order to better measure the mass loss rate of each specimen, the mass of the specimen was weighed every day for three days after water-cooling and the mass before loading, and the corresponding mass loss rate was obtained. The calculating formula is
(1)$$I=\frac{M-M_{w}}{M}$$
where *I* is the mass loss rate, *M* is the weight of specimens before elevated temperatures, and *M_w_* is the weight after exposure to ETWC.

Since the mass loss rate of specimens is mainly related to temperature, various strength grade changes similarly with the temperature, so that with design strength of 60 MPa is taken for comparative analysis. Their variation laws with design parameters are shown in Figure 6, wherein NO.1–NO.4 respectively represent the results obtained after weighing the specimen for the first to fourth time. It can be seen from Figure 6a, when maximum temperatures were less than 400 °C, the mass loss rate of the specimen was smaller, and when maximum temperatures were more than 400 °C, that significantly increased with the rise of temperature. At the same time, when maximum temperatures were less than 600 °C, the mass of each specimen fluctuated within the first three days after the water-cooling. However, when the standing time was longer than three days, the mass of the specimens did not change significantly, indicating that the water content in the specimen was basically stable. For specimens exposed to a temperature of 800 °C, the mass loss rate of specimens was relatively stable, which indicated that the water in the specimen had fully evaporated by the residual heat of the specimen itself after water-cooling. Figure 6b illustrates the variation rule of mass loss rate of each specimen under different cooling methods (the H at the beginning of the specimen No. in the figure represents HSCFST). As can be seen from the figure, the mass loss rate of all the water-cooled specimens was significantly lower than that of the corresponding naturally cooled specimens. According to research by Ke et al. [23], this may be because during the process of water-cooling, water entered the interior and rehydrated with the concrete subjected to high temperatures, forming hydration products that remained some of the water infiltrated into the concrete.

### 3.4. Material Properties after Elevated Temperatures and Water-Cooling

Standard compressive tests of concrete were conducted by three cubic concrete blocks with the size of 150 mm × 150 mm × 150 mm following the GB/T 50152-2012 [24], and measured data are shown in Table 5. According to test data and the analysis of influence factors, the calculation formula of cube compressive strength of high strength concrete block after exposure to ETWC was fitted by introducing the temperature influence coefficient. The constant temperature duration and cooling method are closely related to the temperature field variation of the test block during the test. By considering these two parameters, the influence of compressive strength of high strength concrete during the process of water-cooling can be effectively discussed. However, as the temperature field is a complex problem, more detailed and systematic experiments are needed to study, so in the process of fitting, only the influence of high temperature is considered to calculate the cube compressive strength of high strength concrete after exposure to ETWC. The fitting results are as follows:(2)$$f_{cu}(T)=\left\{ \begin{array}{l} (-0.0005\;T+1.0065\;)f_{cu,20}(20{}^{\circ}C\leq T\leq 400{}^{\circ}C)\;\\ (-0.0014\;T+1.3856\;)f_{cu,20}(400{}^{\circ}C<T\leq 800{}^{\circ}C) \end{array}\right.$$
where *f_cu_*(T) is the cube compressive strength of high strength concrete block after exposure to ETWC, *f_cu_*_,20_ is the cube compressive strength of high strength concrete block under normal conditions and *T* is the maximum temperature of water-cooling blocks. This formula is applicable to the strength estimation of concrete with the strength of 60–80 MPa, subjected to the highest temperature within 800 °C and water-cooling. Figure 7 shows the comparison between the measured compressive strength of the cube test block exposed to ETWC and the results calculated by Equation (2). It can be seen from the figure that the results are in good agreement.

The steel material specimens were cut from the steel tube in the same batch, and they are treated together with HSCFST specimens by ETWC. Mechanical properties indexes were tested according to the GB/T228.1-2010 [25]. The change of steel properties with the maximum temperature is shown in the Figure 8 and the measured value is shown in Table 6. According to the measured steel properties, when the maximum temperature is lower than 400 °C, the strength of steel decreases gradually after water-cooling, the yield strength at 400 °C decreases by 11% compared with that under normal temperature. When the maximum temperature is higher than 400 °C, the strength rises again, the reason is that the steel after ETWC exposure undergoes the effect similar to “quenching,” so the strength shows a slight upward trend.

### 3.5. Loading Procedure

After the formal loading, the slip occurred almost simultaneously at the loading end and free end of the specimen. At the beginning of loading, the slip rate at the loading end of the specimen was faster than that at the free end. With the accumulation of the slip, a “squeak” sound was emitted from the inside of the specimen because of slight concrete crushing. As the damage further aggravated, the sound became clear and the frequency increased. At this time, the push-out load was still in the stage of rapid rise. When the load reached the peak load, the unstable sound of rub-a-dub was given out from the steel tube. With the increase of slippage, the frequency of the sound of rub-a-dub gradually became stable and the sound became louder. At this time, the decreasing rate of the load became slow. During this process, the debris of concrete gradually fell on the steel block at the loading end, and slip lines appeared on the steel tube of some specimens. The slip line extended from the free end to the loading end and developed spirally from top to bottom at a 45-degree angle. When the load remained basically unchanged, the slip at the loading end and free end of the specimen gradually synchronized. After the test, the core concrete at the loading end was concave inward and the steel tube was slightly bulging outward. Obvious slip cracks can be seen between the core concrete and the steel tube. The failure mode of loading end is shown in Figure 9. It can be seen from the figure that the specimen exposed to higher temperature has wider and more obvious slip cracks.

### 3.6. Load-Slip (P-S) Curves

The *P-S* curve of each specimen is shown in Figure 10 (relevant data of HSCFST-22N specimen were not obtained because of operational errors during the loading process).

The development trend of the *P-S* curves of the loading end and the free end of the specimen are basically similar, but at the beginning of loading, the slip of the loading end is more rapid, and the slip of the free end is relatively slow. Approaching the end of the test, the slip of the loading end and the free end is basically synchronous. Damage of specimens gradually accumulated in the push-out process, and the specimen developed from no slip to partial slip and then to overall slip.

The *P-S* curves of specimens can be divided into three sections: upward section, downward section, and gentle section, and its variation can be analyzed from the source of the bonding. In the push-out process of core concrete, the bond force can be composed of three parts, including chemical adhesion force, mechanical interlocking, and friction force on the contact interface. In the initial stage of loading, although the push-out load was very small, the core concrete directly subjected to the load at the loading end had been damaged to a certain extent during the process of elevated temperatures and water-cooling, so it slipped. At this time, the chemical adhesion force provided the main bonding force. With the gradual accumulation of the slip, the chemical adhesion force on the contact interface began to lose. On the one hand, as the concrete was compacted, the mechanical interlocking between the steel tube and the core concrete increased. However, while resisting the push-out action, the bite point of the uneven point inside the core concrete and the steel tube wall was also gradually destroyed, which made its ability to bear the push-out load gradually weaken. On the other hand, the concrete would expand laterally under compression as the push-out load increased, while the circular steel tube can effectively restrain its development and the binding force between the steel tube and concrete would cause friction force between them. During the gradual loss of the force of mechanical interlocking, the friction coefficient on the contact interface would become constant, thus making the friction force become a main source of bond force. During this process, the load curve showed a trend of gradual decline. When the *P-S* curve entered the stable section, there was basically no force of mechanical interlocking, and the main push-out load was borne by the friction force.

As shown in Figure 11, the P-S curve at the loading end of the specimens subjected to ETWC is divided into three types of typical curves, among which the Type I and Type III curve include OA, AB, and BC section, and Type II curve only includes OA and AB section. In particular, the OA section is the upward section, and the three kinds of curves are obviously rising, but the slope of Type I curve is greater than the Type II and Type III curve. For the AB section, this stage is the downward section, the Type I curve decreases linearly in this stage, and this kind of curves mainly appear in the specimens with the exposure temperature less than 400 °C. Section AB and BC of the Type II curve are basically connected, and the slope of section AB is almost zero and in a horizontal state. In this type of curves, the load slightly decreases after passing *P*_u_, and it mainly occurs in the specimen exposed to 400 °C or 600 °C. In the Type III curve, the AB section is the slow descent section, and the slope of the curve decreases gradually during the descent process. The Type III curve appears in the specimen exposed to 800 °C. It should be highlighted that the three kinds of curves fall slowly in BC section and are basically horizontal, so they are treated as gentle sections.

### 3.7. Longitudinal Strain Distribution along Steel Tube

During the push-out test, strain data on longitudinal distribution of specimen along the steel tube can be obtained by attaching the strain gauge to the outer wall of steel tube. Figure 12 presents the strain distribution of some specimens along the longitudinal direction of the steel tube under different load conditions before the load reaches its peak. Where *x* is the distance from the loading end of the specimen to the arrangement point of the strain gauge.

Figure 12 shows that the strain of the outer wall of the steel tube is distributed in an exponential function along the longitudinal direction before the load reaches its peak. It can be fitted by the exponential function constant as *ε*(*x*) = *ae^bx^*. Figure 13 is the fitting results of some specimens, and it can be seen from the figure that the fitting result is good.

In order to analyze the bond action mechanism, the differential method is used to take a micro-segment of steel tube for stress analysis (Figure 14), and the equations of equilibrium is expressed as the following:(3)σ(x)+dσ=τ(x)Sdx+σ(x)
(4)dσ=τ(x)Sdx
(5)$$\tau (x)=\frac{d\sigma }{Sdx}$$
(6)$$d\sigma =E_{s}A_{s}d\varepsilon_{x}$$
when is substituted into Equation (5), then the relationship between bond stress and steel tube strain is obtained:(7)$$\tau (x)=\frac{E_{s}A_{s}}{S}\frac{d\varepsilon_{x}}{dx}$$
when *ε*(*x*) = *ae^bx^* is substituted into Equation (7), then the following is obtained:(8)$$\tau (x)=\frac{abE_{s}A_{s}}{S}e^{bx}$$
where *S* is the perimeter of section of steel tube; *E_s_* is the elastic of steel tube; *A_s_* is the cross-sectional area of steel tube; *τ*(*x*) is the bond stress of steel tube at *x* form the loading end; a and b respectively represent the characteristic values of strain distribution of steel tube.

### 3.8. Characteristic Parameters of P-S Curve

The characteristic parameters of specimens can be obtained from the experimental data, including ultimate bond strength, residual bond strength, and shear bond stiffness. The characteristic parameters of each specimen are shown in Table 7. In the paper, according to the *P-S* curve amount of each specimen, the peak load and corresponding slippage are taken as the ultimate load *P_u_* and peak slippage *S_u_*. *S_r_* is defined as the residual slippage when the slippage reaches 10 mm, while the load matched with *S_r_* is called residual load *P_r_*. At the same time, the secant slope between 0.4*P_u_* corresponding point and coordinate origin at the loading end of the *P-S* curve is taken as the initial shear bond stiffness. The interfacial shear stress related with *P_u_* is defined as the bond strength *τ_u_*, while the interfacial shear stress matched with *P_r_* is called the residual bond strength *τ_r_*; their values can be obtained from Equation (9).
(9)$$\tau =\frac{P}{sl_{a}}$$
where *τ* is the bond strength; *P* is the push-out load, N; *s* is the internal circumference of steel tube, mm; *l_a_* is the anchorage length, mm.

Different countries have different design and criteria for interface bonding strength of concrete filled steel tube. Both DBJ 13-161-2004 (2004) in China and AIJ (1997) in Japan defines the design value of bond strength is 0.225MPa, while BS 5400 (2005) in Britain defines it as 0.4MPa, and EC 4 (2004) in Europe defines it as 0.55MPa [26,27,28,29]. As it can be seen from Table 6, *τ_u_* (1.04~3.80 MPa) and *τ_r_* (0.80~3.02 MPa) of HSCFST subjected to ETWC can still meet the corresponding requirements.

## 4. Factors Influence on Bond Strength

### 4.1. Maximum Temperature

The effect of maximum temperature on ultimate bond strength of specimens subjected to ETWC is presented in Figure 15a. It shows that with increasing temperature, the ultimate bond strength increases first, then decreases, and then increases again. When the maximum temperature is 200 °C, 400 °C, 600 °C, and 800 °C, respectively, the ultimate bond strength is 1.27, 1.55, 1.13, and 1.62 times of that under normal temperature. Thus, it can be seen when the maximum temperature is less than 400 °C, the ultimate bond strength of the specimens increases with temperature rise, but decreases in 600 °C. This is because the interface adhesion between steel tube and concrete mainly includes three parts: chemical adhesion force, mechanical joint force, and frictional force. When the maximum temperature is less than 400 °C, the chemical cementation between them is reduced with the increase of temperature. However, because of the “thermal expansion and cold contraction” effect of different materials, there are differences, especially the concrete materials at elevated temperatures produce irreversible microfracture, which creates deformation in the steel tube that is less than the internal residual deformation of concrete, resulting in the interface to form a pair of compressive force produced by the residual deformation difference after cooling. At a certain temperature range, the strengthening effect of compressive force on the bond strength is greater than the weakening effect of water-cooling on the chemical bond strength. This enhancement effect increases with the rise of temperature, thus making the ultimate bond strength to gradually increase. When the temperature reaches 600 °C, the chemical adhesion force on the interface reduces, and with the loss of concrete strength inside, the mechanical joint force also reduces, and the interfacial bond force is provided mainly by mechanical joint force and frictional force, so the interface ultimate bond strength decreases. However, when the maximum temperature reaches 800 °C, the strength is rebounded. It is different from natural cooling test results; the reasons for this phenomenon is that when the maximum temperature is higher, the rapid drop of temperature will make the deformation of the material increase sharply, which can be proved by the violent crack sound of the internal concrete during the cooling process, which will further increase the compressive force between the interfaces. When the effect of extrusion pressure is greater than that of other factors, it will make the ultimate bond strength of the specimen to rise.

The influence of temperature on the residual bond strength of specimens is illustrated in Figure 15b. With the increase of the maximum temperature, the residual bond strength increases and then decreases, but it is larger than the specimens at normal temperature. When the maximum temperature is 200 °C, 400 °C, 600 °C, and 800 °C, the residual bond strength of specimen is 1.26, 1.64, 1.37, and 1.25 times of that at normal temperature, respectively, among which there is the maximum residual bond strength in 400 °C. For the residual bond strength, it refers to the strength of the specimen at the late slip stage, which is mainly provided by the frictional force. When the maximum temperature is below 400 °C, because of the existence of compressive force between the steel tube and concrete, so that the residual bond strength of specimen gradually increases with the increase of temperature. When the maximum temperature is greater than 400 °C, once the steel tube and concrete form relative sliding, the concrete will be more easily broken after exposure to higher temperature, so it will form a fine particle layer composed of concrete debris, and reduce the friction resistance between the interfaces.

### 4.2. Concrete Strength

The effect of concrete strength on the ultimate bond strength and residual bond strength can be observed from Figure 16. As can be seen from the figure that with the increase of concrete strength, there is no obvious change in them. This is the same as the conclusion in reference [16] that the interfacial bond strength of HSCFST does not change significantly with the increase of concrete strength, and it is also confirmed that the difference between the material properties of high-strength concrete is weakened by the damage after exposure to ETWC.

### 4.3. Constant Temperature Duration

In order to reveal the influence of constant temperature duration on the bond strength of HSCFST specimens subjected to ETWC, the specimens with constant temperature duration of 30 min, 60 min, and 90 min were set respectively, and other influencing factors remained unchanged. Figure 17 shows the influence of constant temperature duration on the ultimate bond strength and residual bond strength of the specimens. It can be seen from the figure that when the constant temperature duration increases from 30 min to 60 min, the ultimate bond strength and residual bond strength of the specimen show a downward trend, which are 7% and 12% lower than that of the specimen with constant temperature for 30 min at 600 °C, and are 7% and 10% lower than that of the specimen with constant temperature for 90 min. It can be seen that the effect of constant temperature duration on the bond strength of the specimen is obvious. In this paper, the bond strength of the specimen decreases gradually with the increase of the duration within 60 min after ETWC exposure. When the constant temperature duration exceeds 60 min, the bond strength between the specimen interface no longer decreases, and basically maintain stability. This shows that when the constant temperature duration reaches 60 min, the core concrete reaches a stable temperature field, and the high temperature damage of concrete is fully developed.

### 4.4. Anchorage Length

In order to study the influence of anchorage length on the bond strength of the specimens subjected to ETWC, two specimens with anchorage length of 280 mm and 430 mm are set up in this paper under the condition of ensuring the same parameters. The influence of anchorage length on bond strength of specimens is shown in Figure 18. It can be seen from the figure that when the anchorage length of the specimen increases, the ultimate bond strength and the residual bond strength between the interfaces decrease. When *l_a_* = 280 mm, the ultimate bond strength and the residual bond strength of the specimen are 2.19 MPa and 1.93 MPa, respectively. Compared with *l_a_* = 280 mm, the ultimate bond strength and the residual bond strength of the specimen *l_a_* = 430 mm decrease by 33% and 27%, respectively, which is consistent with the effect of anchorage length on bond strength of HSCFST subjected to elevated temperature.

### 4.5. Cooling Method

Figure 19 shows the influence of different cooling methods on the bond strength of HSCFST exposed to elevated temperature, in which the relevant data of HSCFST-22N (C80, *l_a_* = 430 mm) specimens are missing. It can be seen from the figure that the ultimate bond strength and residual bond strength of specimens cooled by water spray are lower than those cooled by nature. Compared with the specimens cooled by nature, the ultimate bond strength and residual bond strength of C60, C70, and C80 specimens are reduced by 59%, 52%, 42% and 44%, 40%, 36% respectively. It can be seen from this that spray cooling greatly reduces the bond strength of HSCFST subjected to elevated temperature. According to the phenomenon seen in the spray cooling process, spray cooling causes the temperature of the specimen surface to drop suddenly, which leads to the rapid cold shrinkage of the concrete expanded by high temperatures, making the concrete burst and promoting the development of internal cracks in the concrete, and finally causing severe damage. At the same time, this severe damage also causes the reduction of the bond force between the steel tube and the concrete.

## 5. The Effect of ETWC on Interface Bond Failure

### 5.1. Interface Shear Bond Stiffness

Interface shear bond stiffness is an index to express the ability to resist the sliding deformation caused by shear stress at the interface between the steel tube and high-strength concrete. In this paper, when the load on the *P-S* curve at the loading end is 0.4*P_u_*, the secant slope between the corresponding point and the origin is defined as the elastic shear bond stiffness *K_e_* of the specimen. Figure 20a shows the influence of maximum temperature on the shear bond stiffness of HSCFST specimen subjected to ETWC. Generally speaking, the shear bond stiffness decreases first, then increases, then decreases. When the maximum temperature is 400 °C, the shear bond stiffness of the specimen is the largest, and when the highest temperature is 800 °C, it is the smallest, and 90% of that at normal temperature. Figure 20b reflects the influence of cooling method on the shear bond stiffness of the interface of specimens. It can be seen from the figure that the shear bond stiffness of specimens subjected to water-cooling is smaller than that subjected to natural cooling under different concrete strengths and different anchorage lengths. Therefore, the water-cooling will aggravate the performance degradation of the specimen exposed to elevated temperatures. Compared with specimens cooled by nature, in which the design strength is 60 MPa, 70 MPa, and 80 MPa, the shear bond stiffness of the specimens cooled by water decreases by 4.3%, 30.7%, and 42.7%, respectively. Thus, the decrease range of shear bond stiffness of specimens, which are subject to different cooling method, increases with the increase of concrete strength, and the reason is that the higher the concrete strength is, the more dense the concrete will be, and the more obvious the crack will be after high temperatures and water-cooling, which makes the damage of the concrete more serious.

### 5.2. Interface Energy Dissipation

The energy dissipation factor (*η*) is defined to quantitatively describe energy dissipation capacity during the push-out test. The definition of *η* is as follows:(10)$$\eta =\frac{S_{OHJG}}{S_{OEFG}}$$
where *S_OHJG_* is the shadow area in Figure 21 and *S_OEFG_* is the rectangular area of rectangle OEFG.

Figure 22 reflects the change of the energy dissipation factor of the specimen after ETWC exposure. As can be seen from Figure 22a, when the maximum temperature is less than 600 °C, the energy dissipation of the specimen is not changed much. When the maximum temperature is 200 °C, 400 °C, and 600 °C, respectively, the energy dissipation fluctuates within −2–6% of that of specimen at normal temperature. However, at 800 °C, the energy dissipation of the specimen is reduced by 25% compared with that at normal temperature, with a larger decrease range. This is similar to the law that the interface energy dissipation of HSCFST exposure to elevated temperatures changes with temperature. Figure 22b shows the influence of different cooling methods on the energy dissipation factors of the specimen. From the figure it can be seen that the energy dissipation factors of specimens cooled by water spray are greater than those of specimens cooled by nature, which shows that the energy dissipation capacity of specimens cooled by water spray is better.

## 6. Evaluation of Bond Strength after Exposure to ETWC

The effect of different cooling method on the bond strength of the specimens subjected to elevated temperature is obvious, and the calculation formula in the reference [16] is not suitable for specimens cooled by water. Hence, according to the test data and analysis of influencing factors, with the maximum temperature (*T*), cubic concrete compressive strength (*f_cu_*), and height-to-thickness (*l_a_*/*B*) ratios as the main change parameters, fitting by the least squares method and the obtained formula of the ultimate bond strength and residual bond strength of HSCFST after exposure to ETWC are as follow:(11)$$\tau_{u}=5.64\times 10^{-11}T^{4}-7.9\times 10^{-8}T^{3}+3.11\times 10^{-5}T^{2}-1.93\times 10^{-3}T+0.003825f_{cu}-0.04549\frac{l_{a}}{B}+5.356552$$
(12)$$\tau_{r}=2.73\times 10^{-11}T^{4}-4\times 10^{-8}T^{3}+1.59\times 10^{-5}T^{2}-3.59\times 10^{-4}T+0.017044f_{cu}-0.03093\frac{l_{a}}{B}+2.733574$$
where *T* is the maximum temperature; *f_cu_* is cubic concrete compressive strength; *l_a_* is anchorage length; *B* is thickness of the steel tube. The formula is applicable to the estimation of the bond strength of the HSCFST subjected to ETWC with the concrete strength of C60~C80, the constant temperature duration is about 60 min, and the maximum temperature is within 800 °C.

According to the fitting formula, the comparison between the measured data and the calculated results is given in Table 8. As can be seen from the table, for the ultimate bond strength, the average ratio of *τ_u,t_*/*τ_u,c_* is 0.95 with a standard deviation of 0.104 and a coefficient of variation of 0.11, while for residual bond strength, the average ratio of *τ_r,t_*/*τ_r,c_* is 1.34 with a standard deviation of 0.37 and a coefficient of variation of 0.28. In summary, there is a high consistency between Equations (11) and (12) and the tested results, especially for the ultimate bond strength.

## 7. Conclusions

In this paper, the bond behavior between the steel tube and high-strength concrete after exposure to ETWC was investigated through push-out tests of circular section HSCFST specimens. The following conclusions can be drawn:

(1) After ETWC exposure, the *P-S* curve of the loading end and the free end of the specimen basically has the same changing trend, and the slip development of the loading end is faster than that of the free end. The curves can be roughly divided into three stages: the fast-rising stage, the falling stage, and the gentle stage, and they can be divided into three categories according to their change trend.

(2) The longitudinal strain distribution of the outer surface of the steel tube can be fitted by exponential function, and the fitting effect is good.

(3) With the increase of the maximum temperature, the bond behavior of the specimens exposed to ETWC change is as follows: the ultimate bond strength increases first, then decreases, then increases, while the residual bond strength increases first, then decreases. The shear bond stiffness decreases first, then increases, then decreases. In addition, the interface energy dissipation remains unchanged first, then decreases. All in all, the bond strength of the specimens exposed to ETWC is larger than that of that at normal temperature.

(4) After exposure to ETWC, the bond strength of the specimen does not change significantly with the increase of the concrete strength, and is inversely proportional to the anchorage length. When the constant temperature duration exceeds 60 min, the bond strength of interface no longer decreases with the increase of time, and basically maintain stability.

(5) When the maximum temperature is 600 °C, the ultimate bond strength, residual bond strength, and the bond shear stiffness of water-cooling specimens are smaller than those of natural cooling specimens, but the interface energy dissipation is higher.

(6) According to the measured data and the analysis results of the influencing factors, the calculation formula of the bond strength of HSCFST specimens subjected to ETWC is obtained, and the calculated value is in good agreement with the test value.

## Figures and Tables

**Figure 1 materials-13-00150-f001:**
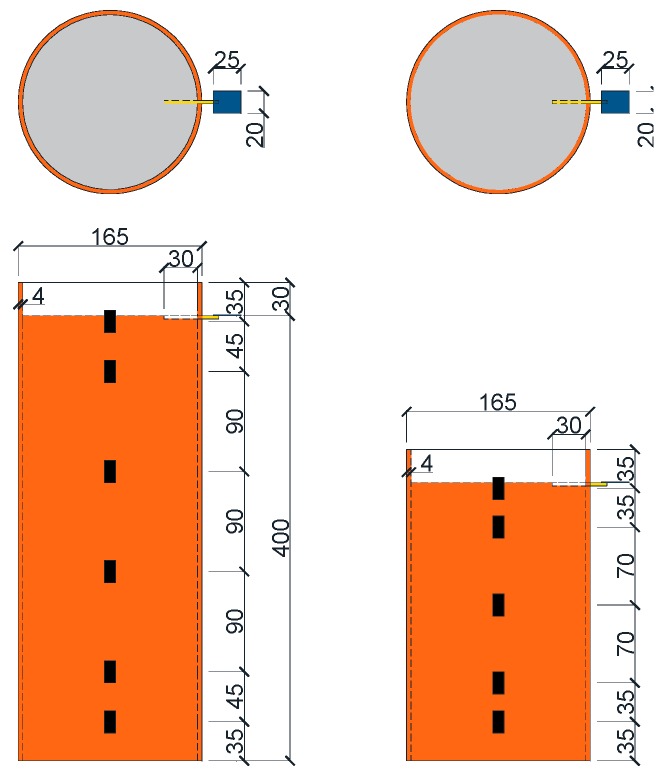
Section size and strain gauges distribution of specimens (mm).

**Figure 2 materials-13-00150-f002:**
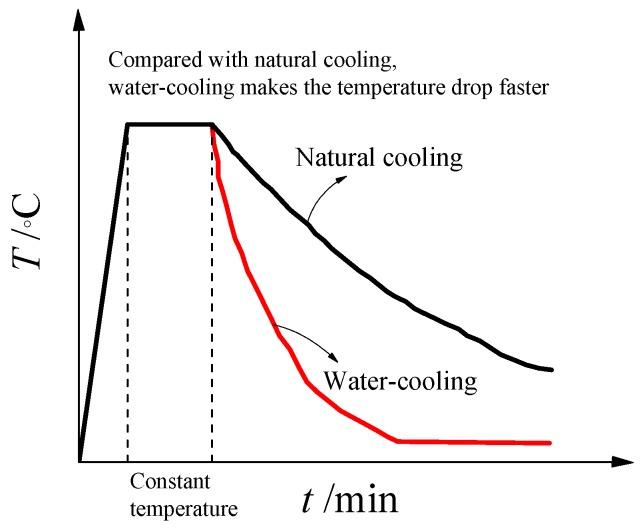
Schematic diagram of heating.

**Figure 3 materials-13-00150-f003:**
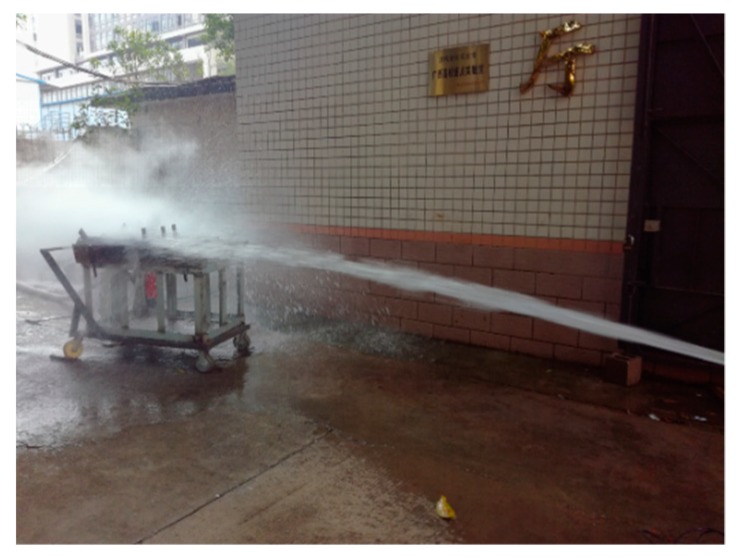
Specimens cooled by fire hydrant after high temperatures exposure.

**Figure 4 materials-13-00150-f004:**
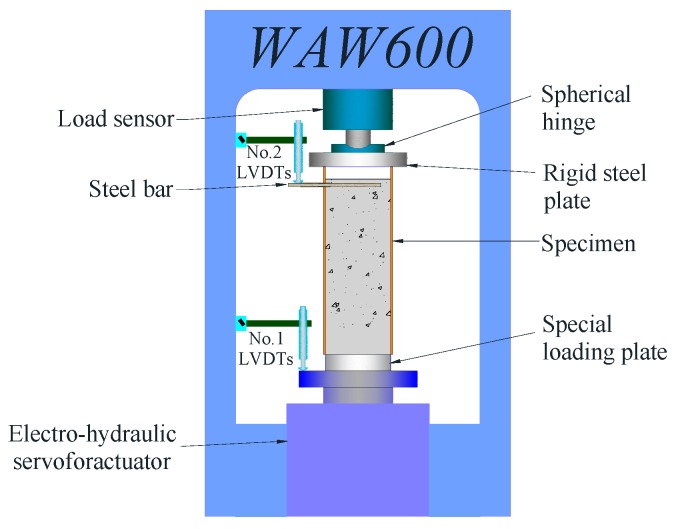
Push-out test setup.

**Figure 5 materials-13-00150-f005:**
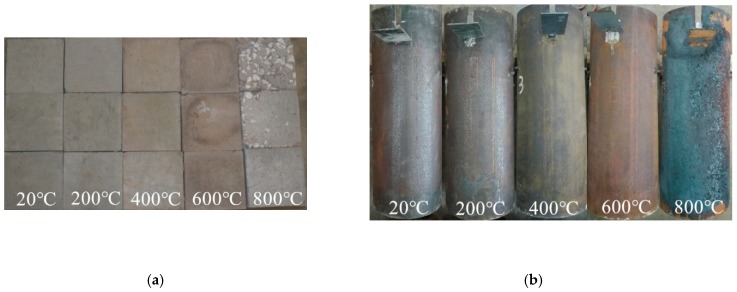
Appearance of specimens after water-cooling. (**a**) high strength concrete test blocks; (**b**) specimens.

**Figure 6 materials-13-00150-f006:**
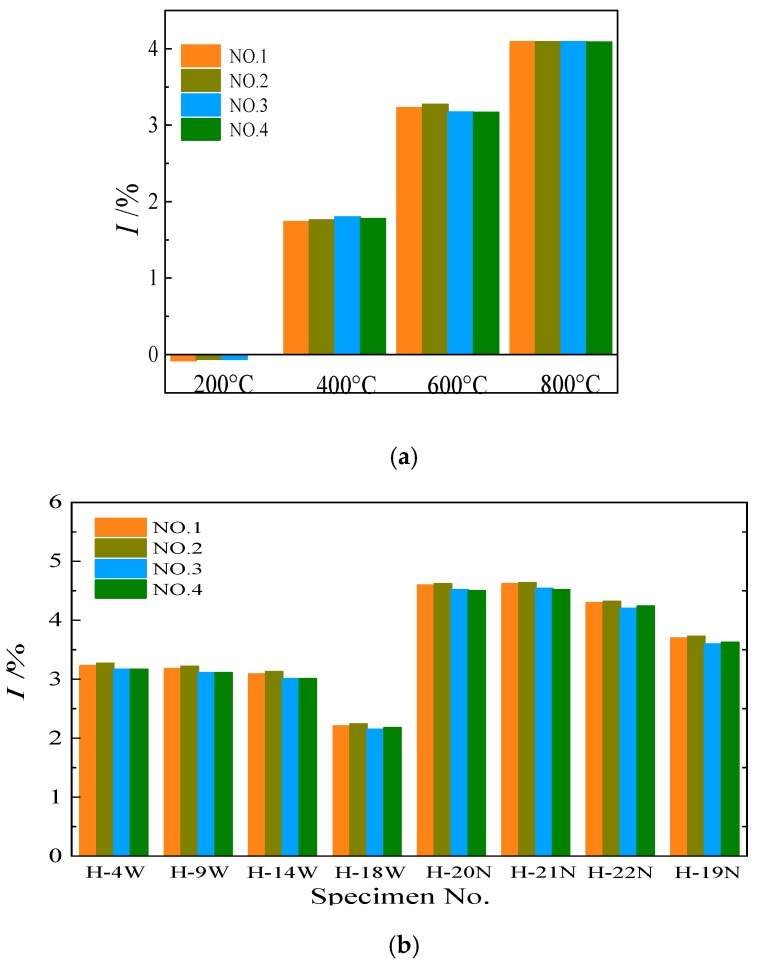
The mass loss rate of specimens exposed to water-cooling (**a**) the effect of temperature (60 MPa); (**b**) the effect of cooling methods (600 °C).

**Figure 7 materials-13-00150-f007:**
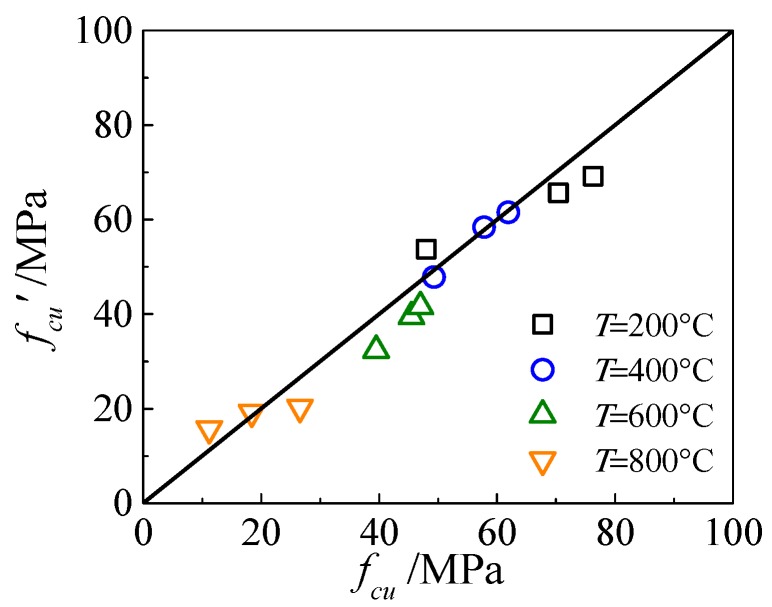
Comparison of test and calculation values.

**Figure 8 materials-13-00150-f008:**
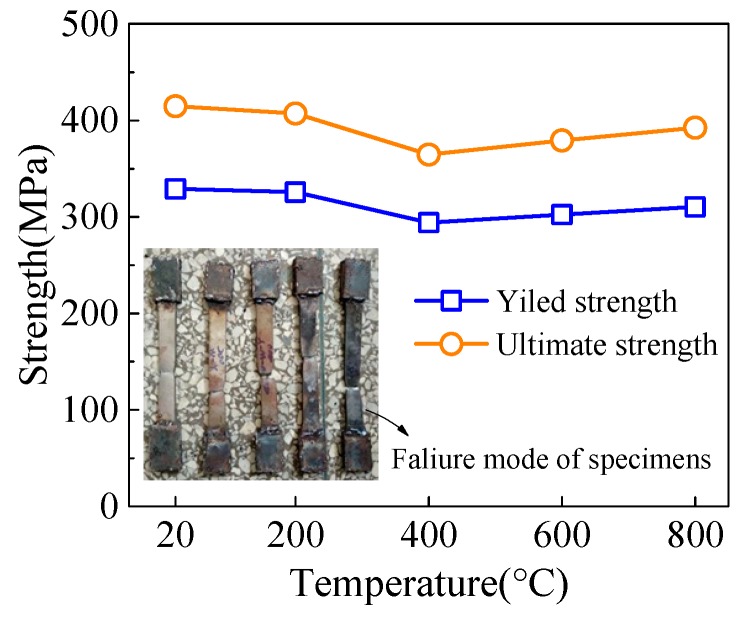
Measured steel properties.

**Figure 9 materials-13-00150-f009:**
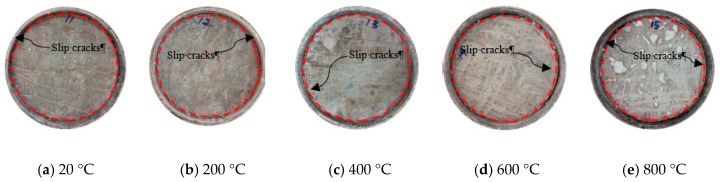
Failure pattern of the loading end after push-out test.

**Figure 10 materials-13-00150-f010:**
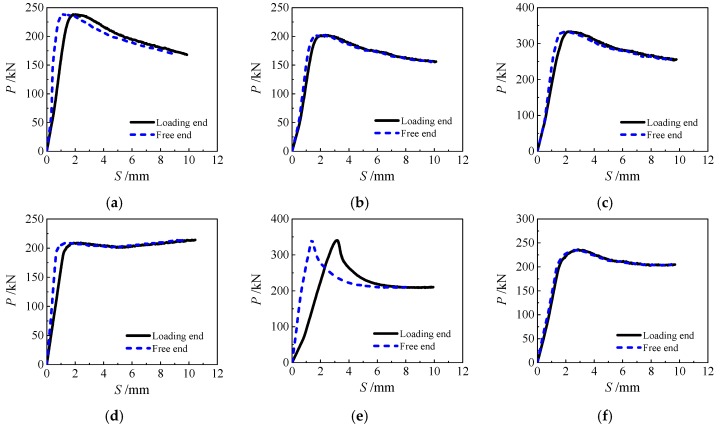
Load-slip curves of specimens (**a**) HSCFST-1W; (**b**) HSCFST-2W; (**c**) HSCFST-3W; (**d**) HSCFST-4W; (**e**) HSCFST-5W; (**f**) HSCFST-6W; (**g**) HSCFST-7W; (**h**) HSCFST-8W; (**i**) HSCFST-9W; (**j**) HSCFST-10W; (**k**) HSCFST-11W; (**l**) HSCFST-12W; (**m**) HSCFST-13W; (**n**) HSCFST-14W; (**o**) HSCFST-15W; (**p**) HSCFST-16W; (**q**) HSCFST-17W; (**r**) HSCFST-18W; (**s**) HSCFST-19W; (**t**) HSCFST-20W; (**u**) HSCFST-21W.

**Figure 11 materials-13-00150-f011:**
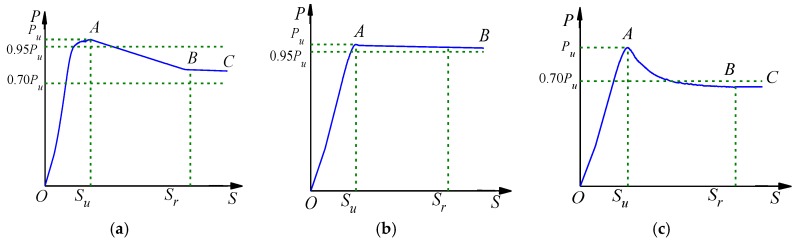
Typical curve classification (**a**) I; (**b**) II; (**c**) III.

**Figure 12 materials-13-00150-f012:**
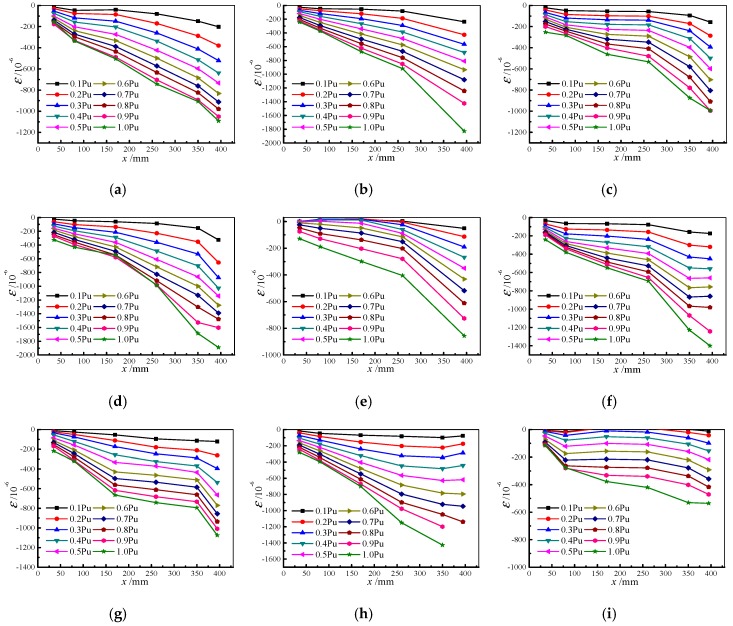
Strain distributed along longitudinal direction in some specimens (**a**) HSCFST-1W; (**b**) HSCFST-2W; (**c**) HSCFST-4W; (**d**) HSCFST-6W; (**e**) HSCFST-7W; (**f**) HSCFST-9W; (**g**) HSCFST-11W; (**h**) HSCFST-12W; (**i**) HSCFST-14W.

**Figure 13 materials-13-00150-f013:**
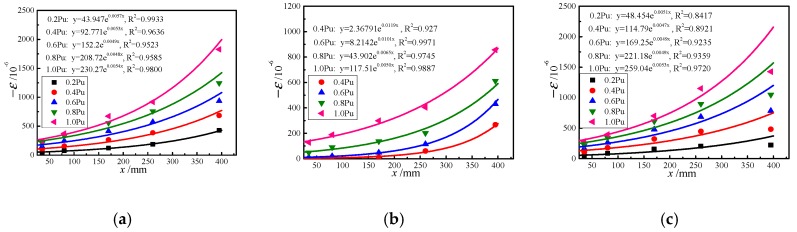
Fitting results of strain distribution along longitudinal direction (**a**) HSCFST-2W; (**b**) HSCFST-7W; (**c**) HSCFST-12W.

**Figure 14 materials-13-00150-f014:**
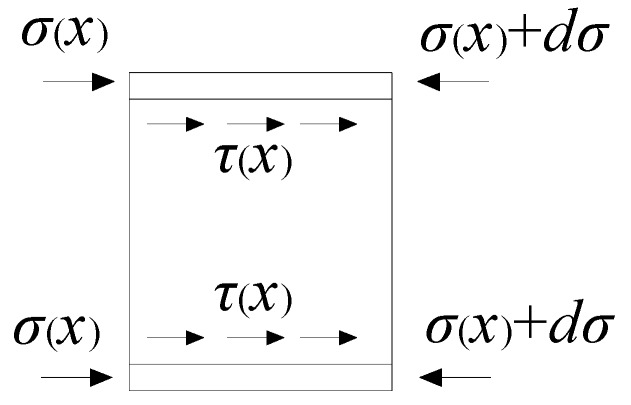
Stress distribution along the micro steel.

**Figure 15 materials-13-00150-f015:**
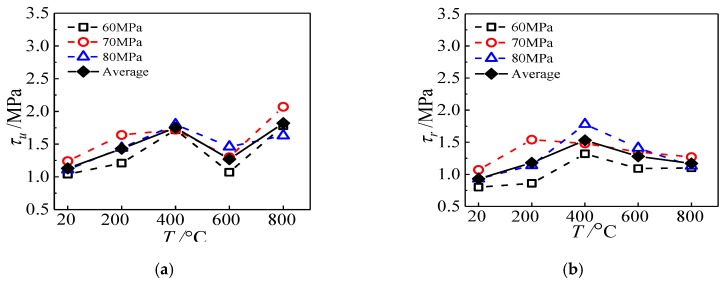
The effect of maximum temperature on bond strength (**a**) ultimate bond strength; (**b**) residual bond strength.

**Figure 16 materials-13-00150-f016:**
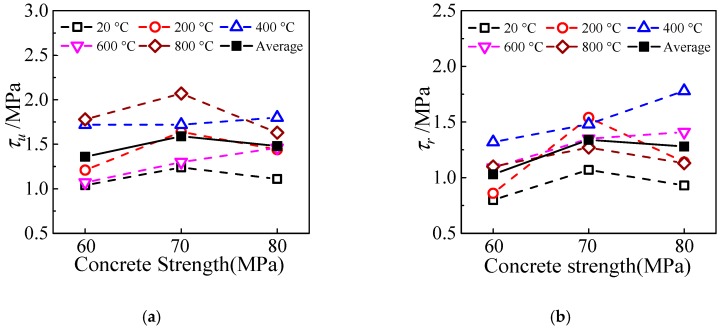
The effect of concrete strength on bond strength (**a**) ultimate bond strength; (**b**) residual bond strength.

**Figure 17 materials-13-00150-f017:**
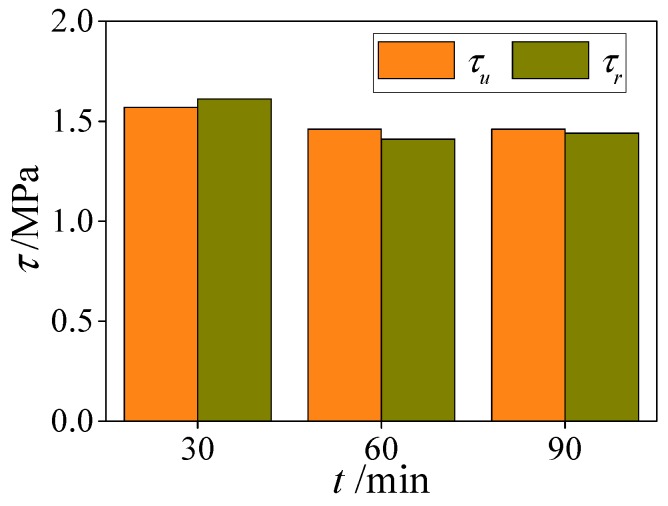
The effect of constant temperature duration on bond behaviors.

**Figure 18 materials-13-00150-f018:**
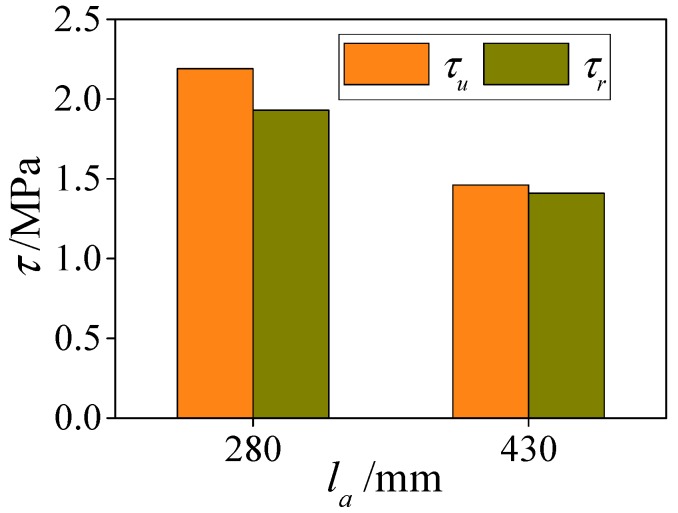
The effect of interface length on bond behaviors.

**Figure 19 materials-13-00150-f019:**
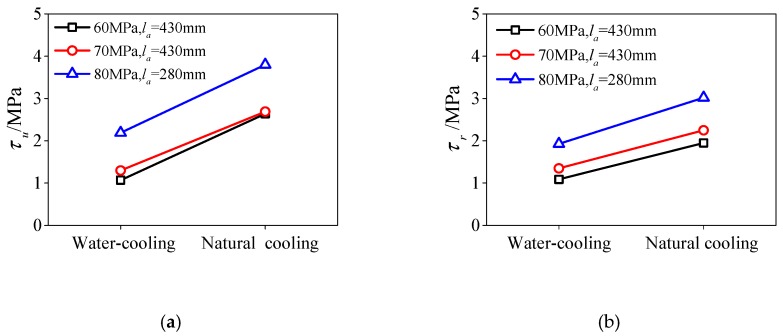
The effect of cooling method on bond behaviors (**a**) ultimate bond strength; (**b**) residual bond strength.

**Figure 20 materials-13-00150-f020:**
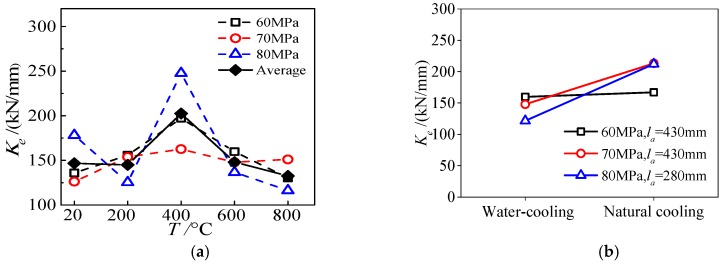
Shear bond stiffness of specimens (**a**) maximum temperature; (**b**) cooling method.

**Figure 21 materials-13-00150-f021:**
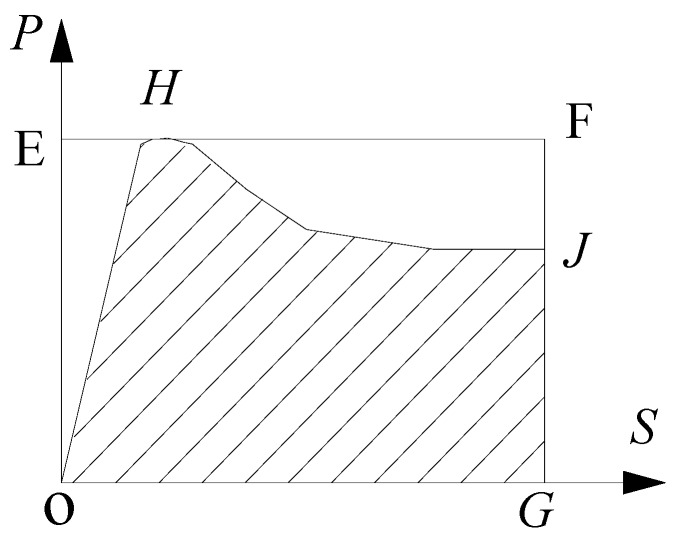
Energy dissipation area model.

**Figure 22 materials-13-00150-f022:**
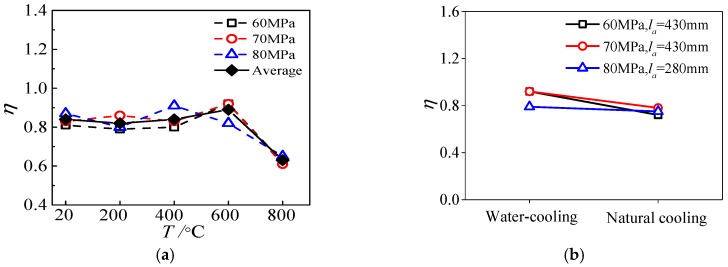
Energy dissipation factor (**a**) maximum temperature; (**b**) cooling method.

**Table 1 materials-13-00150-t001:** Characteristics of specimens.

Identification of Specimen	Design Value of Concrete Strength	*T*	*t*	*H*	Cooling Method	*l_a_*
		(°C)	(min)	(mm)		(mm)
HSCFST-1W	60 MPa	20	60	430	water-cooling	395.67
HSCFST-2W		200	60	430	water-cooling	398.33
HSCFST-3W		400	60	430	water-cooling	392.33
HSCFST-4W		600	60	430	water-cooling	398.33
HSCFST-5W		800	60	430	water-cooling	388.67
HSCFST-6W	70 MPa	20	60	430	water-cooling	386.33
HSCFST-7W		200	60	430	water-cooling	388.50
HSCFST-8W		400	60	430	water-cooling	376.17
HSCFST-9W		600	60	430	water-cooling	384.00
HSCFST-10W		800	60	430	water-cooling	397.67
HSCFST-11W	80 MPa	20	60	430	water-cooling	394.33
HSCFST-12W		200	60	430	water-cooling	394.67
HSCFST-13W		400	60	430	water-cooling	391.50
HSCFST-14W		600	60	430	water-cooling	390.33
HSCFST-15W		800	60	430	water-cooling	386.33
HSCFST-16W	80 MPa	600	30	430	water-cooling	386.00
HSCFST-17W		600	90	430	water-cooling	399.88
HSCFST-18W	80 MPa	600	60	280	water-cooling	242.33
HSCFST-19N		600	60	280	natural cooling	245.67
HSCFST-20N	60 MPa	600	60	430	natural cooling	387.00
HSCFST-21N	70 MPa	600	60	430	natural cooling	389.67
HSCFST-22N	80 MPa	600	60	430	natural cooling	387.17

**Table 2 materials-13-00150-t002:** Concrete proportions.

Design of Value Concrete Strength	Material Content /(kg/m^3^)						
	Cement	Sand	Gravel	Water	Fly Ash	Silica Fume	Water Reducer
60 MPa	400	600	1280	165	60	4.0	5
70 MPa	450	680	1128	160	70	5.0	7
80 MPa	520	628	1117	155	80	5.5	9

**Table 3 materials-13-00150-t003:** The physical properties of coarse aggregate.

Crushing Index (%)	Apparent Density (kg/m^3^)	Loose Bulk Density (kg/m^3^)	Compact Bulk Density (kg/m^3^)	Loose Porosity (%)	Compact Porosity (%)	Water Absorption (%)	Water Content (%)
12.5	2702	1520	1650	43.74	38.93	0.373	0.013

**Table 4 materials-13-00150-t004:** The physical properties of sand.

Fineness Modulus	Apparent Density (kg/m^3^)	Loose Bulk Density (kg/m^3^)	Compact Bulk Density (kg/m^3^)	Loose Porosity (%)	Compact Porosity (%)	Water Absorption (%)	Water Content (%)
3.02	2637	1579.2	1689.5	40.11	35.93	0.654	0.285

**Table 5 materials-13-00150-t005:** Concrete strength exposed to elevated temperatures and water-cooling (ETWC).

Design Value of Concrete Strength	Cube Compressive Strength under Different Temperature (MPa)				
	20 °C	200 °C	400 °C	600 °C	800 °C
60 MPa	59.3	48.0	49.3	39.5	11.2
70 MPa	72.4	70.4	57.8	45.5	18.4
80 MPa	76.3	67.9	61.9	47.0	26.6

**Table 6 materials-13-00150-t006:** Steel properties exposed to ETWC.

Strength Index	Maximum Temperatures				
	20 °C	200 °C	400 °C	600 °C	800 °C
Yield strength *f*_y_, (MPa)	329.1	325.7	294.1	302.5	310.4
Ultimate strength *f*_u_, (MPa)	414.6	407.1	364.6	379.3	392.4

**Table 7 materials-13-00150-t007:** Characteristic values of test results.

Specimen No.	*P_u_*/kN	*S_u_*/mm	*P_r_*/kN	*S_r_*/mm	*τ_u_*/MPa	*τ_r_*/MPa
HSCFST-1W	202	2.51	156	10.13	1.04	0.80
HSCFST-2W	238	1.86	168	9.86	1.21	0.86
HSCFST-3W	333	2.30	256	9.77	1.72	1.32
HSCFST-4W	209	2.40	214	10.44	1.07	1.09
HSCFST-5W	340	3.19	210	9.93	1.78	1.10
HSCFST-6W	236	2.87	205	9.67	1.24	1.07
HSCFST-7W	314	3.24	295	9.95	1.64	1.54
HSCFST-8W	318	2.40	273	10.17	1.72	1.48
HSCFST-9W	246	3.23	255	9.95	1.30	1.35
HSCFST-10W	406	3.07	250	9.83	2.07	1.27
HSCFST-11W	214	2.85	180	9.78	1.11	0.93
HSCFST-12W	280	3.11	220	10.69	1.44	1.14
HSCFST-13W	347	3.30	344	9.85	1.80	1.78
HSCFST-14W	280	2.34	271	10.10	1.46	1.41
HSCFST-15W	310	2.98	216	10.13	1.63	1.13
HSCFST-16W	299	2.92	307	10.09	1.57	1.61
HSCFST-17W	287	2.82	284	9.93	1.46	1.44
HSCFST-18W	261	2.09	230	9.44	2.19	1.93
HSCFST-19N	460	2.56	366	10.00	3.80	3.02
HSCFST-20N HSCFST-21N	503 517	4.25 3.84	353 429	12.74 11.50	2.64 2.69	1.95 2.25

**Table 8 materials-13-00150-t008:** Comparison between calculations and experimental values.

Specimen No.	Ultimate Bond Strength			Residual Bond Strength		
	*τ_u,t_*	*τ_u,c_*	*τ_u,t_*/*τ_u,c_*	*τ_r,t_*	*τ_r,c_*	*τ_r,t_*/*τ_r,c_*
HSCFST-1W	1.04	1.05	0.98	0.8	0.69	1.16
HSCFST-2W	1.21	1.33	0.92	0.86	0.76	1.13
HSCFST-3W	1.72	1.68	1.03	1.32	1.08	1.23
HSCFST-4W	1.07	1.26	0.84	1.09	0.73	1.48
HSCFST-5W	1.78	1.99	0.89	1.1	0.51	2.15
HSCFST-6W	1.24	1.21	1.02	1.07	0.98	1.1
HSCFST-7W	1.64	1.52	1.07	1.54	1.22	1.26
HSCFST-8W	1.72	1.89	0.91	1.48	1.35	1.1
HSCFST-9W	1.3	1.45	0.9	1.35	0.95	1.42
HSCFST-10W	2.07	1.92	1.08	1.27	0.56	2.26
HSCFST-11W	1.11	1.13	0.97	0.93	0.99	0.94
HSCFST-12W	1.44	1.44	1	1.14	1.13	1.01
HSCFST-13W	1.8	1.73	1.04	1.78	1.3	1.37
HSCFST-14W	1.46	1.38	1.06	1.41	0.92	1.52
HSCFST-15W	1.63	2.08	0.78	1.13	0.79	1.43
HSCFST-18W	2.19	3.06	0.71	1.93	2.07	0.93

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
