# Peer review of "Interfacial Bond Behavior of High Strength Concrete Filled Steel Tube after Exposure to Elevated Temperatures and Cooled by Fire Hydrant"

_materials, 2019, doi:10.3390/ma13010150_

Round 1

Reviewer 1 Report

General remark: The article is an original contribution and the temperature-dependent bond behavior of high strength concrete filled steel tubes. The article is well structured. However, the language style and grammar should be checked to ensure a better readability. Further, the following remarks might be addressed to improve the quality of the manuscript.

Introduction: Please provide an overview / description about the content of the manuscript. It may help the reader to know what to expect.

Experimental program: Line 104: Specify the superplasticizier used (what do you mean by concrete mixing station?)., Line 105: Use aggregates instead of stones., Table 3 + 4: Please defined the different types of porosity mentioned here., Figure 3: not needed because the reader should know how an oven looks like and the figure does not provide any information. Figure 4: also not needed. Line 156: Define the preload. In general, this section is not complete because several further tests are conducted which are not mentioned here.

Experimental results / Analysis / Evaluation: 163 ff: How do you distinguish between the water from the spraying and the water vapor produced afterwards? How do you determine the amount of water vapor? Line 183: Please give possible explanations for the color changes. Line 188: Why do the test blocks and the specimens show different colors? Line 202: Is it realistic that hydration restarted due to water from the water-cooling? Did you find evidence for that? Line 207: The description of the experimental methodology should be rather included in Section 2. Line 222: didn´t --> did not. Line 213 ff: Also refer to other microstructural changes in concrete during the exposure to elevated temperatures (it is not only the water that is responsible for changes of the mass and the mechanical behavior). Line 229: Provide evidence for this statement. Line 250: The procedure of how these equations were derived are not clear. Line 339: Defined Type I curve a.s.o. Line 344: Fig. 14 is --> Fig. 14 presents (also for other sentences). Line 375: What are the variables a and b? Line 407 ff: The interpretation in the following is mainly based on the chemical adhesion force, mechanical joint force and frictional force. However, there is no experimental evidence for their occurrence and interaction given. Please refer to this and explain in more detail the reasons why you assigned the temperature-dependent phenomena to these effects. Line 447: Word is missing in this sentence.

Author Response

Point 1: Line 104: Specify the superplasticizier used (what do you mean by concrete mixing station?).

Response 1: There is a problem with the line 104 expression, change it to high range polycarboxylate water reducer from concrete mixing station was used.

Point 2: Line 105: Use aggregates instead of stones.

Response 2: Coarse aggregates has been used instead of stones.

Point 3: Table 3+4: Please defined the different types of porosity mentioned here.

Response 3: In Table3 and Table4, the loose porosity and the compact porosity are based on the loose bulk density and the compact bulk density of the material respectively, which are specified in Chinese standards GB/T14684-2011 and GB/T14685-2011.

Point 4: Figure 3: not needed because the reader should know how an oven looks like and the figure does not provide any information. Figure 4: also not needed.

Response 4: According to the expert's opinion, the author has deleted Figure 3 and Figure 4.

Point 5: Line 156: Define the preload. In general, this section is not complete because several further tests are conducted which are not mentioned here.

Response 5: In order to eliminate the contact error of the LVDTs, the load shall be applied to 10% of the estimated ultimate load before the formal loading of each test, and then the unloading shall be carried out to ensure the normal operation of the instrument. It should be emphasized that some common material performance tests, such as compressive strength test of concrete blocks and tensile test of steel, are not described in detail by the author, but relevant standards are provided for reference in the paper.

Point 6: How do you distinguish between the water from the spraying and the water vapor produced afterwards? How do you determine the amount of water vapor?

Response 6: Because the spray water will only produce a small amount of water mist, which can be ignored compared with the amount of water vapor generated during water spraying, and the water vapor is evaporated upward, so it is easy to distinguish the two. At the same time, the amount of water vapor is significantly different produced by the specimen subjected to water-cooling after different thermal exposure, so it can be directly observed.

Point 7: Line 183: Please give possible explanations for the color changes. Line 188: Why do the test blocks and the specimens show different colors?

Response 7: According to the difference of the test phenomena, the author thinks the ETWC exposure causes the color changes in the test blocks. Because the outer layer of the specimen is a steel tube, the appearance change of the specimen is different from that of the test block after exposure to ETWC.

Point 8: Line 202: Is it realistic that hydration restarted due to water from the water-cooling? Did you find evidence for that?

Response 8: Please review the following work: http://dx.doi.org/10.3969/j.issn.1007-9629.2017.05.023, it proves that hydration will happen again due to water from the water-cooling.

Point 9: Line 250: The procedure of how these equations were derived are not clear.

Response 9: This equation is obtained by computer fitting based on the measured data.

Point 10: What are the variables a and b?

Response 10: a and b respectively represent the characteristic values of strain distribution of steel tube.

Point 11: Line 407 ff: The interpretation in the following is mainly based on the chemical adhesion force, mechanical joint force and frictional force. However, there is no experimental evidence for their occurrence and interaction given. Please refer to this and explain in more detail the reasons why you assigned the temperature-dependent phenomena to these effects.

Response 11: Because the existence of these interactions between interfaces depends on the behaviours of materials that make up the interface, especially the material behaviours of concrete, and after the high temperature action, the behaviours of concrete will appear serious degradation, which will inevitably lead to the change of the interaction between interfaces. Take into account this, the author has carried out this part of analysis.

Some minor problems have been reasonably modified according to the expert's suggestions, and the author will not point out one by one here. Thank you again for your valuable suggestions

Reviewer 2 Report

The authors present a paper on concrete filled steel tubes. The theme presented is interesting and very topical. The work in general is well organized. The planning is correct and the analysis and interpretation of results is coherent and well structured. The conclusions presented are based on the results obtained but could be more assertive and present limit or target values or trends.
In my opinion this work can be considered for publication.

Author Response

Point 1: The authors present a paper on concrete filled steel tubes. The theme presented is interesting and very topical. The work in general is well organized. The planning is correct and the analysis and interpretation of results is coherent and well structured. The conclusions presented are based on the results obtained but could be more assertive and present limit or target values or trends. In my opinion this work can be considered for publication.

Response 1: Thanks for your comments and suggestions, I will make improvements based on your suggestions in the next work.

Reviewer 3 Report

This paper presents findings from an experimental program that examined the effect of elevated temperature and cooling regime on the bond-slip behavior in concrete filled steel tubes. The findings of this work are interesting and this work presents a number of useful illustrations. The following items are to be addressed before this work is publishable:

Why did the author use “fire hydrant” in the title? What are the main differences between using fire hydrant and traditional water used in firefighting applications? Is water in fire hydrant any different (i.e. chemical composition etc.)? What was the rate of cooling applied in natural and controlled cooling of the specimens? Does formula no. 2 work for all types of concrete tested here (60. 70 and 80MPa)? Did any of the concrete heated specimen spall under fire? Does the quenching effect continue beyond 400C? this seems a bit odd as compared to work published in literature. These work show that steel might have a slight increase in strength under elevated temperature which goes back to degrade once it passes 400-600C. It is rare to see steel continue to undergo increase in strength beyond 400-600C. please review some of the following works: https://doi.org/10.1016/j.conbuildmat.2019.04.182 https://doi.org/10.1016/0379-7112(88)90032-X There seems to be a distinct difference between Type I, Type II and Type III curves. Please discuss details on fundamental differences in these figures. Formulas 11 and 12 are a bit complex/long. Could a more simple form be provided?

Author Response

Point 1: Why did the author use “fire hydrant” in the title? What are the main differences between using fire hydrant and traditional water used in firefighting applications? Is water in fire hydrant any different (i.e. chemical composition etc.)?

Response 1: Why the author use “fire hydrant” in the title, the reason is that compared with traditional water used in firefighting applications, the water from the fire hydrant has a large water pressure. When it is sprayed to the surface of the structure subjected to high temperature, it will cause more violent physical and chemical changes. In terms of composition, there is not any between the two.

Point 2: What was the rate of cooling applied in natural and controlled cooling of the specimens?

Response 2: For the specimens treated by water-cooling, the spray duration shall be uniformly controlled at 25 minutes, and the spray head shall be the same, and then the specimens shall be placed at the same place. The specific cooling rate was not measured by the tester. In later studies, researchers will accept suggestions to measure the cooling rate to make up for the deficiency.

Point 3: Does formula no. 2 work for all types of concrete tested here (60. 70 and 80MPa)?

Response 3: Yes, it can be seen from Figure 9 that the calculated value of the formula is in good agreement with the test value.

Point 4: Did any of the concrete heated specimen spall under fire? Does the quenching effect continue beyond 400°C?

Response 4: First of all, after the high temperature test, there is no observable phenomenon of concrete peeling, but the surface concrete falls off due to the impact of water flow after water-cooling. Secondly, according to the article provided by the experts, the strength of the steel is indeed reduced after exposure to elevated temperature of 400 degrees, because they do not have the quenching effect, which requires water cooling to achieve.

Point 5: There seems to be a distinct difference between Type I, Type II and Type III curves. Please discuss details on fundamental differences in these figures.

Response 5: As shown in Figure 13, the P-S curve of at loading end of specimens subjected to ETWC is divided into three types of typical curves, among which the Type I and Type III curve include OA, AB and BC section, the Type II curve only includes OA and AB section. In particular, the OA section is the upward section, and the three kinds of curves are obviously rising, but the slope of Type I curve is greater than the Type II and Type III curve. For the AB section, this stage is the downward section, the Type I curve decreases linearly in this stage, and this kind of curves mainly appear in the specimens with the exposure temperature less than 400°C. Section AB and BC of the Type II curve are basically connected, and the slope of section AB is almost zero and in a horizontal state. In this type of curves, the load slightly decreases after passing Pu, and it mainly occurs in the specimen exposed to 400°C or 600°C. In the Type III curve, the AB section is the slow descent section, and the slope of the curve decreases gradually during the descent process. The Type III curve appears in the specimen exposed to 800°C. It should be highlighted that the three kinds of curves fall slowly in BC section and are basically horizontal, so they are treated as gentle sections.

Point 6: Formulas 11 and 12 are a bit complex/long. Could a more simple form be provided?

Response 6: This equation is obtained by computer fitting based on the measured data. In order to maintain the accuracy of the calculation, the author does not suggest modifying it.

Round 2

Reviewer 1 Report

The reviewers´ remarks have been addressed properly so that the quality of the manuscript has been improved. The experimental design is described more clearly and some language mistakes have been corrected.

Reviewer 3 Report

Thank you for your efforts.